# Selecting and analysing climate change adaptation measures at six research sites across Europe

Henk-Jan van Alphen[1], Clemens Strehl[2], Fabian Vollmer[2], Eduard Interwies[3], Anasha Petersen[3], Stefan Görlitz[3], Luca Locatelli[4], Montse Martinez Puentes[4], Maria Guerrero Hidalga[5], Elias Giannakis[6], Teun Spek[7], Marc Scheibel[8], Erle Kristvik[9], Fernanda Rocha[10], Emmy Bergsma[1]

[1]KWR Water Research Institute, Nieuwegein, 3433 PE, The Netherlands
[2]IWW Water Centre, Mülheim an der Ruhr, 45476, Germany
[3]InterSus - Sustainability Services, Berlin, 10405, Germany
[4]Aquatec (Suez Spain), 08038 Barcelona, Spain
[5]CETAQUA Water Technology Centre, 08940 Barcelona, Spain
[6]The Cyprus Institute, Energy Environment and Water Research Center, Nicosia, 2121, Cyprus
[7]Province of Gelderland, Arnhem, 6811CG, The Netherlands
[8]Wupperverband, Wuppertal, 42289, Germany
[9]Norwegian University of Science and Technology, NO-7491 Trondheim, Norway
[10]Laboratório Nacional de Engenharia Civil, 1700-066 Lisboa, Portugal

*Correspondence to*: Henk-Jan van Alphen (henk-jan.van.alphen@kwrwater.nl)

**Abstract**

As Europe is faced with increasing droughts and extreme precipitation, countries are taking measures to adapt to these changes. It is challenging, however, to navigate through the wide range of possible measures, taking into account the efficacy, economic impact and social justice aspects of these measures, as well as the governance requirements for implementing them. This article presents the approach of selecting and analysing adaptation measures to increasing extreme weather events caused by ongoing climate change that was developed and applied in the H2020 project BINGO (Bringing Innovation to Ongoing Water Management). The purpose of this project is (a) to develop an integrated participatory approach for selecting and evaluating adaptation measures, (b) to apply and evaluate the approach across six case-study river basins across Europe, and (c) to support decision-making towards adaptation capturing the diversity, the different circumstances and challenges river basins face across Europe. It combines three analyses: governance, socio-economic and social justice The governance analysis focuses on the requirements associated with the measures and the extent to which these requirements are met at the research sites. The socio-economic impact focuses on the efficacy of the measures in reducing the risks and the broad range of tools available to compare the measures on their societal impact. Finally, a tentative social justice analysis focuses on the distributive impacts of the adaptation measures. In the summary of results, we give an overview of the outcome of the different analyses. In the conclusion, we briefly assess the main pros and cons of the different analyses that were conducted. The main conclusion is that although the research sites were very different in both the challenges and the institutional context, the approach presented here yielded decision relevant outcomes.

# 1    Introduction

Along the process of adapting to climate change, finding and defining appropriate adaptation measures is a complex task.

Moreover, it is the key activity to increase the resilience to future climate change induced risks (Dogulu and Kentel, 2015). In addition, good practice in selecting adaptation measures is a fundamental task in adjusting water infrastructure to climate change, which is globally needed (Wilby, 2019). Part of this good practice is to analyse the impact of potential adaptation measures, not only in terms of hazard risk reduction, but also in terms of socio-economic effects, social justice or governance needs for implementation. For example, Zhou et al. (2012) combine climate modelling and an economic cost-benefit

assessment in analysing climate adaptation measures for pluvial flooding in urban areas. Harrison et al. (2013) combine climate change scenarios with socio-economic scenarios in a digital platform to allow stakeholders to explore adaptation options within the context of varying futures. European research projects such as ECONADAPT and BASE have also focused on the economics of climate adaptation to support adaptation planning (Watkiss et al. 2015, Garotte et al. 2016, Meyer et al. 2015)).

Another part of this good practice is to involve stakeholders in selecting and analysing these adaptation measures. Involving local stakeholders in these analysis, not just through consultation, but through co-production, enhances their relevance, usability, legitimacy and credibility (Palutikof et al. 2019). For example, Bhave et al. (2014) combine top-down climate modelling with bottom-up (involving stakeholders) prioritization of adaptation measures, but do not analyse socio-economic effect of measures, nor governance requirements for implementation. Andersson-Sköld et al. (2015) use focus group interviews

with stakeholders to gauge the perceptions of adaptation measures, as part of a broader integrated framework to analyse the impact of climate adaptation measures. On the other hand, Singh et al. (2020) develop and apply a broad framework to assess the feasibility of adaptation measures, including political, economic and social indicators, but not specific to local conditions and not as part of a participatory framework. This study contributes to the literature by integrating three different analyses (governance, socio-economic, and social justice) in a participatory framework, where most other studies capture only one or

two of the above-mentioned dimensions (Verkerk et al. 2017; Bojovic et al., 2018; O'Sullivan et al., 2020).

This article presents the approach of selecting and analysing adaptation measures to increasing extreme weather events caused by ongoing climate change that was developed and applied in the H2020 project BINGO (Bringing Innovation to Ongoing Water Management). The purpose of this project is (a) to develop an integrated participatory approach for selecting and

evaluating adaptation measures, (b) to apply and evaluate the approach across six case-study river basins across Europe, and (c) to support decision-making towards adaptation capturing the diversity, the different circumstances and challenges river basins face across Europe. The project was conducted by over 20 project partners at six research sites in Europe: (1) The city of Badalona (Spain), which faces the risk of flash floods and combined sewer overflows (CSOs) due to increased precipitation; (2) The city of Bergen (Norway), also facing the risk of floods and CSOs due to increased precipitation; (3) The Veluwe

(Netherlands), a Natura 2000 site where long term drought may affect the groundwater system; (4) the Troodos mountains

(Cyprus), where decreasing precipitation causes water shortages for farmers and communities; (5) The Wupper River Basin, which is divided in two sub cases, one about flood risk due extreme weather events and one about decreasing water levels in the main water reservoir due to decreasing precipitation; and (6) the Sorraia Valley (Portugal) where farmers are confronted with water shortages due to decreasing precipitation.


The BINGO project followed a comprehensive approach from decadal predictions of weather events, hydrological analysis of the impact of the weather events on water systems, to risk analysis and risk treatment. The work presented in this article focuses on the treatment of risks following extreme precipitation or drought. Risk treatment in project BINGO was organised as a collaborative process between scientists and local stakeholders, through Communities of Practice (CoPs) (Freitas et al., 2018).

These CoPs consisted of representatives of local and regional governments, organisations involved in climate adaptation and research partners. CoPs provide a social context in which researchers and stakeholders can engage in formal and informal interactions and co-analyse and co-produce the contextual knowledge that is necessary for climate change adaptation (Iyalomhe et al. 2013). The CoPs in the BINGO project were locally created and externally supported by the scientific project partners, which is found to be necessary condition for a sustainable CoP (Vincent et al, 2018)


Based on the risks that were identified and analysed in the risk analysis, the CoPs selected and analysed adaptation measures, with the goal of informing decision makers about the expected efforts and gains from the implementation of these measures. The approach applied in the BINGO project is in line with steps formulated in the Adaptation Support Tool developed as part of the Climate-ADAPT initiative of the European Union (https://climate-adapt.eea.europa.eu/knowledge/tools/adaptation-

support-tool). More resources from the BINGO project can be found on the project website (www.projectbingo.eu).

The next sections describe the process of selecting and analysing promising adaptation measures in the order as conducted within the BINGO project for all cases: (1) collecting and selecting adaptation measures, (2) governance analysis of selected adaptation measures (3) analysis of socio-economic implications (4) social justice analysis. These steps are illustrated with

examples from the case study in the city of Badalona as well as from other sites in brief. A summary of the results of the analysis is provided, comparing different types of measures. Finally, conclusions are drawn on the application of the different methods.

## 2 Collecting and selecting adaptation measures

Two approaches were applied to collect potential adaptation measures suitable to the climate change risks identified at the six

research sites, namely a desk study of previous adaptation research and consultation of stakeholders involved in the local CoPs. For the desk study, the primary sources for adaptation measures were two previous EU research projects CarpathCC (http://www.carpathcc.eu/) and PREPARED (http://www.prepared-fp7.eu). From both projects databases were available with

adaptation measures, including a brief analysis of their potential impact and risk reduction potential. From these databases the BINGO research partners selected measures that were (a) potentially relevant for the hazards the research sites are facing and

(b) relevant for the main characteristics of the research site (e.g. urban area, agricultural area, natural area). At the same time, in each of the six research sites the first CoP meeting was organised. In this meeting, local stakeholders discussed and identified potential future climate hazards for their research site and identified measures that were either already planned or considered suitable.

These measures were collected as part of workshop reports (Van Alphen et al., 2017a) and compiled, together with the

measures from the desk research that were selected by the research sites. In total, 91 measures were collected. In many cases, research sites reported similar measures with slightly different wording, or very specific measures could be placed in a broader category. Through this reduction, 44 measures were compiled in a portfolio of adaptation measures (Van Alphen et al., 2017b). The Portfolio of Adaptation Measures is now available as an online tool (http://beta.tools.watershare.eu/bingo/$/). In the portfolio, different types of measures are distinguished. Informational measures (e.g., raising awareness for behavioural

change), financial measures (e.g., insurance and subsidies), regulatory measures (e.g., standards and legal bans) and infrastructural measures (e.g., flood control infrastructure). The complete set of measures can be filtered by type of risk, sector, or adaptation objective. Since the portfolio was first created to support the work in BINGO, the broad risk categories reflected the risks first identified in the six case studies (1) decrease of water quantity due to decrease precipitation; (2) decrease of water quality due to decreased precipitation; (3) floods due to increased precipitation; (4) decrease of water quality due to

increased precipitation. The sectors reflected the sectors represented in the case studies: (1) agriculture; (2) flood management; (3) public water supply; (4) urban drainage (5) water governance (6) water resource management. This design was chosen so project partners and future users can easily find measures suited to their own circumstances. For each measure an analysis of the governance needs for implementation was given, based on the Three Layer Framework presented below. This analysis was done by research partners and was not based on specific conditions at the research sites, but on desk research.

After compiling this broad portfolio, a more specific assessment of potential risks at the research site was made and discussed with stakeholders. Local stakeholders could make a selection of adaptation measures from the longlist provided by the project team and the measures that were developed locally. This first selection of measures was accompanied by a discussion on the following governance aspects related to the measures: (1) responsibility for implementation, (2) participation/division of roles, (3) availability of necessary resources; (4) potential challenges. During the CoP meetings at the six research sites, these aspects

were discussed for the different measures and a selection was made either through scoring or through voting. The measures were selected for the purpose of further analysis. For instance, in the case of Cyprus, measures were first scored on relevance and feasibility and then voted on by the stakeholders. In some cases, stakeholders decided to analyse measures that were not part of the portfolio, but came up in the stakeholder process after the portfolio was already compiled. Table 1 shows the selection of measures for each research site.

**Table 1: Overview of adaptation measures selected by the research sites**

| Research site – climate risk | Technical infrastructure measures | Blue/green measures | Behavioral measures | Socio-economic analysis applied (also see figure 1) |
|---|---|---|---|---|
| *Wupper River Basin, Germany* – Insufficient reservoir storage due to drought | Water transport between reservoir catchments Alternative water source (horizontal well) | | Water Saving Reduction of low water elevation | Cost Effectiveness Analysis (CEA) |
| Flood risk due to increased precipitation | Technical protection measures for property Alignment protection Retention Basin | | | CEA with Multi Criteria Analysis (MCA) |
| *Veluwe, The Netherlands* Decreasing ground water levels due to drought | Artificial infiltration | Land use change (pine to broadleaf) | Agricultural water restrictions | MCA |
| *Sorraia Valley (Tagus basin), Portugal* Decreasing ground water levels due to drought | Rehabilitation and modernization of irrigation networks | | Tagus water resources management model | CEA |
| *Troodos, Cyprus* Constraints on public water supply and irrigation due to drought | Desalination Use of treated sewage water for irrigation Maintenance of groundwater recharge systems Irrigation scheduling technologies | | | CEA with MCA |
| *Bergen, Norway* Combined Sewer Overflow due to increased precipitation | Sewer separation Safe Flood Ways | Sustainable Urban Drainage Systems (SUDS) | | CEA |
| *Badalona, Spain* Combined Sewer Overflow and Flash Floods due to increased precipitation | Increase of sewer capacity | SUDS | Early Warning System | Cost Benefit Analysis (CBA) |

## 3  Governance analysis of selected adaptation measures

### 3.1 Three Layer Framework

The Three Layer Framework for Water Governance, a tool for assessing water governance practices (Havekes et al., 2016), was used to analyse the governance needs of the adaptation measures. The framework builds on the work done by the Organization for Economic Co-operation and Development (OECD 2011) on governance gaps in water governance, and elaborates on these gaps with building blocks for good water governance identified by the Dutch Water Governance Centre. The framework distinguishes between three layers of governance: the *content layer*, the *institutional layer* and the *relational layer*. First, the *content layer* looks into the substance of adaptation measures. Measures are characterized by the risk that they address (such as from floods, CSOs or droughts) and the type of intervention (informational, financial, regulatory, infrastructural). Also, the content layer addresses the type of knowledge and expertise needed to implement the measure (technical knowledge, administrative knowledge, knowledge about interest and preferences). Second, the *institutional layer* deals with the broad range of organizational requirements for the implementation of adaptation measures. This entails: (1) the involvement of the necessary actors and a clear division of roles and responsibilities between them; (2) the administrative resources to implement the measure, such as staff, accounting and monitoring capacities, regulatory capacity and knowledge infrastructure; (3) the legal requirements and the connection with EU regulation, policy and directives; and finally (4) the financial requirements and the way these funds can be generated. Third, the *relational layer* of the framework refers to the requirements placed on the wider governance context of adaptation to climate change. This entails: (1) the potential cultural or ethical issues that may support or obstruct implementation of adaptation measures; (2) the requirements with regard to public accountability, communication and participation.

Based on this Three Layer Framework, a questionnaire was developed to assess each individual measure selected by the CoPs. The questions address the different layers and their elements. Examples of questions are: which (constellation of) actors should be involved in the development and implementation of the adaptation measure? Are the necessary actors currently involved sufficiently? Which cultural or ethical issues either support or obstruct the implementation of the adaptation measure? The questionnaires were filled in by the research partners or in a collaborative effort with experts and local stakeholders.

### 3.2 Application in the Badalona case

Following the methodology outlined above, three adaptation measures were selected for the Badalona research site with the objective of reducing of urban floods and CSOs or reducing the impact thereof. These include: conventional urban drainage grey infrastructure (e.g., new or larger drainage conduits, new detention tanks, new surface drains, etc.); the development of SUDS and the implementation of an Early Warning System (EWS).

For each one of the adaptation measures a governance assessment was performed by following the expert analysis of the three-layer-framework. The results of the analysis demonstrate that: (1) the structural measure (increase of sewer capacity) meets the knowledge and legal requirements (this measure was already included in the Drainage Master Plan Badalona of 2012) but does not have the financial, organizational and relational requirements for its implementation; (2) the SUDS development meets the technical and relational requirements (it has quite support given it is a "green solution") but does not meet the financial, legal and organizational requirements to foster its implementation; (3) the Early Warning System meets almost all the requirements except from the relational layer regarding public accountability, communication and participation.

This governance assessment (together with the socio-economic assessment explained next) has allowed the Badalona City Council to have a clear roadmap to support decisions towards urban adaptation.

## 4 Analysis of socio-economic implications

### 4.1 Guidance in selecting fitting analysis frameworks

To achieve a viable adaptation to climate change is a complex task that is highly dependent on factors such as the financial means of involved stakeholders and the social impacts of the implementation of a measure. For decision makers it is key to define all necessary indicators and acquire the necessary data for the evaluation. Guidance is needed to find the framework that best fits the specific case, depending on the need to include e.g. not only monetary but also non-monetary decision indicators (Markanday et al., 2019; Dogulu and Kentel, 2015).

Within the BINGO project a toolbox was compiled that summarizes the state-of-the art of suitable methods for evaluating and comparing alternative strategies and measures for climate change adaptation (Koti et al., 2017). This toolbox has been used as a background framework to analyse and prioritize fitting risk reduction measures for the six research sites, customized to local stakeholders' needs. The work conducted in the BINGO project resulted in the preparation of a decision tree that supports stakeholders to identify suitable assessment methods, respectively depending on their requirements and preferences to the analysis process. Complementing the comprehensive BINGO-toolbox, the decision tree in fig. 1 focuses on those analysis frameworks applied in the BINGO case studies.

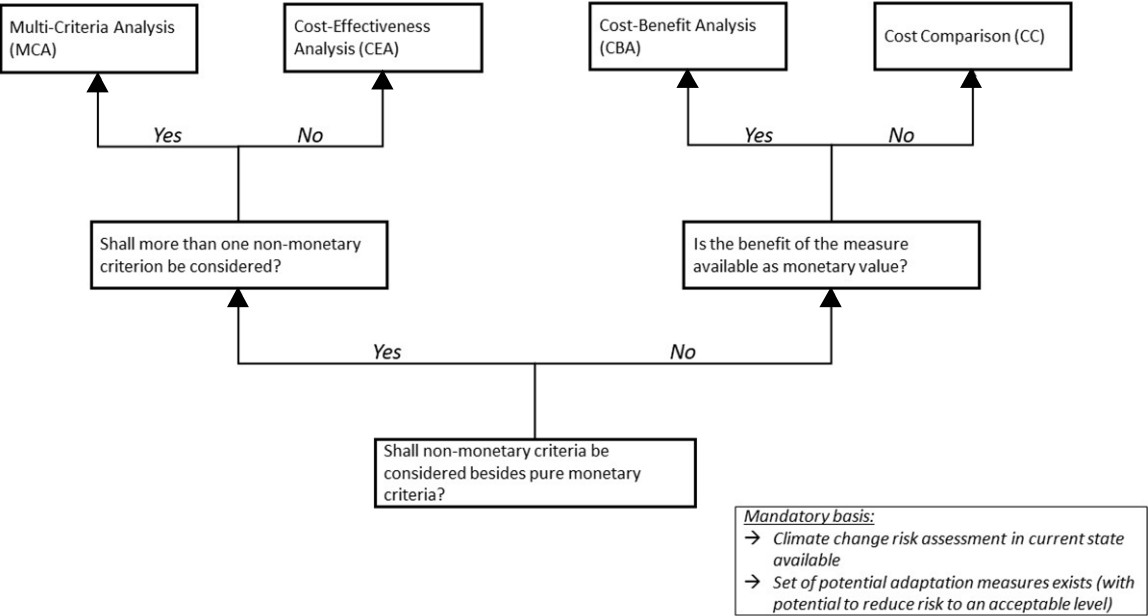


**Figure 1: Decision tree supporting the definition process of a fitting analytical framework to evaluate socio-economic implications of climate change adaptation measures**

The application of the decision tree presupposes the definition of potential adaptation measures. This is due to the fact that the provided methods aim to support the analyst in prioritizing a set of potential adaptation measures. The work conducted in the

BINGO case studies showed that the nature of potential adaptation measures (e.g. infrastructural measures, behavioural measures, etc.) can have a major influence on the requirements of the analysis methods and relevant indicators, underlying the need for a case specific analysis method. Furthermore, a risk assessment of expected climate change hazards and their magnitude needs to be conducted before hand. This is important in formulating a base line (expected future without any adaptation measure). In this way the risk reduction potential compared to that base line can be assessed for all alternative

adaptation measures, in order to evaluate the potential risk reduction of each measure. This is a mandatory data set to compare alternative adaptation measures with one of the methods presented in the framework above. This risk reduction potential should be used as a primary indicator. For example, in the application of a cost-effectiveness analysis (CEA) it can serve as input to compare the costs to the risk reduction effectiveness.

In selecting an evaluation framework by using the decision tree, the participation of all stakeholders that are affected by the

adaptation measures turned out to be of high importance. These stakeholders might be water authorities, local or regional governments, NGOs, farmers, or local residents. The BINGO case studies showed the importance of stakeholders getting the chance to express their points of view and major concerns. This holistic integration of stakeholder perspectives enabled the

definition of sets of indicators for prioritization of adaptation measures and that ensured the eventual acceptance of the results by all stakeholders. An omission of this broad stakeholder participation might lead to a lack of stakeholders' acceptance of the analysis results and thus to major barriers in the implementation of the adaptation measures. Limitations in the final choice of an evaluation framework may arise due to insufficient data availability, e.g. because required data does not exist or because the efforts to get the required data is incommensurate with the benefits gained.

The following sections briefly highlight why and in which case studies of the BINGO project the decision support frameworks have been applied. This is not a comprehensive presentation of the results, since this would exceed limits of this article. Details can be found throughout the documentation in BINGO project reports (http://www.projectbingo.eu/resources).

## 4.2 Cost-Benefit Analysis

A CBA helps to obtain a rank of available options in monetary terms. It is a commonly used approach to prioritize flood risk reduction measures for climate change adaptation (Penning-Rowsell et al. 2010, Zhou et al. 2012). Costs represent the resources necessary to implement a certain measure. In this context, benefits account for the expected reduction of monetary damages brought by the measures implementation. In addition, co-benefits can be included for measures that improve ecosystem services provision, such as green infrastructure, which are evaluated in monetary terms by available valuation methods (OECD 2018, Gerner et al. 2018, Hanley and Barbier 2009).

A CBA was conducted for the Badalona case study, due to suitability with the data available and general interest among stakeholders. The costs of the measures under assessment contain: (1) initial investments, included gradually in a linear trend following the assumptions of future implementation times, (2) operating costs for the time horizon of the analysis (set until 2100), (3) rehabilitation and disposal costs, considering technical assumptions on the duration of the assets.

Benefits were assessed using the avoided cost methods, consisting of the estimation of the difference between estimated damages in the baseline scenario and in each of the alternative scenarios. Expected Annual Damage (EAD) was used as an indicator (Martinez-Gomariz et al., 2019) for flood damages, calculated for Badalona using historical flood damage data provided by the National Reinsurance Consortium (Consorcio de Compensación de Seguros). In addition, for the green roof and other green areas proposed as measures, ecosystem service benefits were identified as regulating (air quality and temperature control), supporting (habitat creation), and cultural (aesthetic) services. Monetization of the changes on the environmental variables were estimated using market prices for the marketed items (e.g. reduction of electricity consumption from temperature control), and also non-market prices for those items that do not have a market for trade (e.g. increase of property value after green roof implementation). For non-market prices, benefit transfer method has been applied, using reference studies and adapting the values in economic and size terms. For more details on the methodologies and results, please refer to the deliverable D5.3 of the project (Strehl et al. 2019a).

## 4.3 Cost-Effectiveness Analysis

The core idea of a CEA is to relate the costs of a measure to its effectiveness, like the technical performance (Levin and McEwan, 2001). Both key figures, the costs as well as the effectiveness, which is measured with a suitable indicator, need to be quantified to calculate the ratio. Within BINGO, a CEA was used in the case study of the Große Dhünn reservoir (Wupper River Basin). The reservoir, operated and owned by the Wupperverband (regional water board), usually stores up to 81 Mm³ of water used for drinking water production, supplying up to 1M people. In this case the risk assessment conducted in the project pointed out the potentially hazardous event of more than 1,000 days with an insufficient reservoir water storage (defined as less than 35 Mm³ water storage) in the worst case decadal climate change projections. Therefore, the focus of this case study was to explore infrastructural and non-infrastructural adaptation measures that reduce the risk to an acceptable level.

In this particular case, effectiveness was measured by a non-monetary indicator, namely its technical performance which was defined as the additional amount of available water per year. The Wupperverband had the capacity to simulate the additional amount of water based on the reduction of the low water elevation (non-infrastructural measure) and by a transfer pipeline from the so called Kerspe reservoir to the Große Dhünn reservoir (infrastructural measure). Moreover, the additional water availability by a new horizontal well (infrastructural measure) and by water saving devices coupled with water use restrictions as emergency action (non-infrastructural measure) could be estimated. The data availability allowed a cost estimation for all four measures. Thus, a cost-effectiveness analysis was the best fitting decision support method in this case, offering the possibility to rank technically and/or organizationally feasible risk reduction measures by their cost-effectiveness ratio, advising the Wupperverband and other regional stakeholders in the prioritization of climate change adaptations for their regional situation. More details can be found in Strehl et al. (2019a).

## 4.4 Multi-Criteria Analysis

An MCA describes a class of analysis methods that consider a variety of different criteria (synonym: indicators) to achieve a prioritization of the potential measures. A common application is the weighted sum method. Here, first the stakeholders affected by the potential adaptation measures have to agree on a set of relevant indicators to evaluate the impacts of the different measures. Afterwards the stakeholders have to give a weight to each indicator. In the subsequent step each indicator is evaluated by the stakeholders with respect to its manifestation for each respective measure, e.g. by applying a scale from 1 (negative manifestation) to 5 (positive manifestation). Finally, the score for each measure is determined by summing up the products of the weighting and the evaluation score of each measure. These final scores serve as ranking of the measures (Carrico et al. 2014).

This method was applied in the Veluwe case study. The Veluwe is a region in the Netherlands dealing with hazards of long-term droughts and warming/heat stress. To reduce the risks connected to these hazards, three potential adaptation measures were identified, namely the reduction of areas covered by pine-trees, the implementation of artificial surface water infiltration

and agricultural water restrictions. As a separate cost-effectiveness analysis was conducted in the Veluwe case, an MCA was
chosen as second decision support that focused on 19 different non-monetary indicators that the group of relevant stakeholders
agreed on. This methodology enabled a focused investigation of the manifestation of different non-monetary indicator besides
the cost-effectiveness analysis, allowing to take a well-founded and holistic decision for or against the respective adaptation
measures (Strehl et al. 2019a).

## 4.5 Cost Comparison

Cost comparison (CC) is a dynamic approach used to compare the costs. Investment expenditures as well as operational
expenditures for implementing and operating an adaptation measure are accounted for along the lifetime of a measure, also
minding discounting (Götze et al. 2015, DWA 2012). The advantage of a CC in general is that it allows a straightforward
comparison of adaptation measures by one single common indicator. Thus, this method is a viable approach to support decision
making in climate change adaptation if only cost data is available for potential adaptation measures, or if the costs are the most
important indicator and other indicators are negligible.

Within the BINGO case studies, no solely CC was conducted as the data availabilities in all case studies allowed a more
complex analysis, incorporating more than one single indicator for decision support analysis. However, the underlying
methodology for a CC was used in many of the case studies, e.g. in the case study for the Große Dhünn reservoir (Wupper
River Basin) to calculate the annual costs for adaptation measures.

## 4.6 Combining frameworks

The decision tree explained above serves as a guidance that is suitable for a variety of cases where decisions for or against
certain adaptation measures need to be taken. However, sometimes a combination of analysis frameworks might be necessary
or desired. Within the BINGO project, this was essential for the case study of the Wupper River Basin. The spatial boundaries
of that case study covered an area of approx. 8 km² around a small urban water course called the Mirke creek. The area is
known as endangered flood zone (MKULNV 2015) and recent flood damage events triggered the urgency of involved
stakeholders to act since flood risk might also aggravate with further climate change in the future. The aim of the case study
was to compare potential flood risk reduction measures at several so called critical hotspots along a 6 km long course of the
creek. The explored measures needed to be ranked by their cost-effectiveness, in order to advise stakeholders where to spend
time and financial resources first (Strehl et al. 2019b).

To capture all relevant socio-economic details, the customized approach for Wuppertal had to combine some of the frameworks
mentioned in fig 1 above. In the Wupper River Basin case, stakeholders stated from the beginning of the project that non-
monetary indicators are also relevant for this case study. However, as stated above, the primary aim was to rank the solutions

in order to guide stakeholders how to spend time and financial resources wisely, beginning at a hotspot with the best cost-effectiveness. This is why a CEA was combined with an MCA framework.

The MCA framework followed in the Wupper River Basin case study was aligned to the so called Analytical Hierarchy Process (AHP) based on Saaty (2008) and Saaty (1987). Here, at first a weighing of the indicators was given by the stakeholders by pairwise comparisons of the indicators, followed by an evaluation of the indicators' manifestations themselves. Both values per indicator were afterwards combined to a final value that indicates the respective measure's effectiveness in non-monetary terms. The resulting single value was related to the costs for each measure (as calculated by the principles of a CC). Details on
the followed approach and results of the case can be found in the BINGO D5.3 report (Strehl et al. 2019a).

## 5 Social justice analysis

### 5.1 Why a social justice analysis?

Social justice and equity principles have been highlighted by the IPCC (2018) as key aspects of a climate-resilient development of societies. Adaptation to climate change is difficult to regulate because the causes and effects of a changing climate are
spread both geographically and in time. For policy-making on climate adaptation to be legitimate and effective, it has to take justice and equity principles into account (Gupta 2005, Caney 2005b). Adaptation policies also contribute to human well-being and social capital, and increase the overall adaptive capacity of societies (Reckien et al. 2018).

Until today, the debate on social justice and climate change has mainly centred on the recognition of responsibility for global climate change (Pielke et al. 2007), inter-generational justice (Caney 2005a) as well as distributional justice, especially in the
context of vulnerability to impacts of climate change (Adger 2006, Breil et al. 2018). It is only recently that social justice is emerging as a central concept to guide decision making for adaptation policy. In the face of climate change, the scope of the transition ahead calls for a high degree of support from all parts of society. The successful implementation of adaptation action thus depends on transparent and legitimate decision making processes as well as a systematic consideration of equity principles (Patterson et al. 2018). A social justice analysis of adaptation measures, especially with an advanced methodology to introduce
the topic into adaptation decision making, has great potential to assess the probable acceptability of proposed measures, to inform their context-adequate design and to enhance the legitimacy of the planning process with a view to the long-term support by the wider public.

### 5.2 The concept of social justice in BINGO

There is not a commonly agreed definition of social justice or equity in the context of adaptation (Breil et al. 2018), and the
prioritization of principles and values varies according to the specific regional context (EEA 2018). In essence, social justice theorizes about fair allocations of burdens and benefits among different members of a society (Rawls 1971). According to Miller (1999) social justice thus concerns the question of "how the basic structure of a society distributes advantages and

disadvantages to its members". These distributions are often based on, and legitimized through, "distributive" or "equity"
principles (Buchanan 1972, Cook 1987). The BINGO social justice analysis seeks to map the distributions of costs or negative
impacts and benefits of the adaptation measures among different actors or groups in society in the specific context of each
research site. This was done using a standardized questionnaire (see fig. 2). Participants also received a short introduction
paper, highlighting the concept of social justice to them as well. The questionnaire was developed based on three equity
principles generally distinguished in the environmental-philosophical literature (Shue 1999, Low and Gleeson 1998, Paavola
& Adger 2002, Ikeme 2003, Anand 2004): (1) the egalitarian principle is based on Mill's and Benthams' utilitarian "greatest
happiness principle". Distributions aim to maximize the positive effects and minimize the negative effects for society as a
whole. An example of this principle in adaptation governance are the upcoming international weather insurances and bonds,
which pay out after a certain weather disaster irrespective of the needs of the victims (Dlugolecki & Keykhah 2002); (2) the
solidarity principle aims to neutralize "involuntary inequalities" between people. Distributions follow Rawls' "maximin"
principle which involves maximizing the well-being of those who are worst-off. A practical example of the operation of this
principle in adaptation governance is the United Nations Adaptation Fund that finances adaptation projects in developing
countries (Person & Remling 2014); (3) the deontological principle is based on Kant's notion that people are rational and act
intentionally, and can therefore be held responsible for their choices and actions. Nozick's elaborated on this notion in his
"entitlement theory", which holds that any "patterned" redistributions focused on outcomes are unjust and (re)distributions
should always put individual rights and liberties at the basis. The "polluter pays principle" is a practical example of this
principle (Tol & Verheyen 2004).

As the evaluation of social justice is highly context dependent, the analysis does not present a conclusive result for each
measure but rather presents a qualitative summary of distributional impacts for decision makers to consider in addition to the
rating which is produced in the socio-economic assessment.

Figure 2: Questionnaire for social justice analysis.

**5.3 The application of social justice analysis in BINGO - the Badalona case study**

In the BINGO case study of Badalona, the application of the social justice analysis for the three selected adaptation measures shows that (1) all adaptation measures will have positive impacts on Badalona's citizens. The general public will benefit from the reduction of flooding and combined sewer overflows and the social perception in the municipality's efficiency will increase; (2) none of the adaptation measures are likely to incur negative side-effects; on the contrary, the implementation of nature-based solutions will incur social co-benefits such as: enhanced public amenity, enhanced air quality, increase of ecosystem services and the reduction of the "heat island effect"; (3) regarding equity principles, both the deontological and egalitarian principles may apply in the case of climate change adaptation given that, on the one hand, Badalona's citizens are paying for the proper performance of the urban drainage system and at the same time the society as a whole receives the positive consequences of such adaptation.

**5.4 Limitations**

Pre-existing inequalities or specific vulnerabilities of certain groups of the respective municipalities could only be considered to a limited extent (question 7 of the questionnaire). However, the analysis of specific social vulnerabilities at the level of the municipality is advisable when designing adaptation measures as well as the participation of vulnerable groups in the planning process to ensure that the contextual and procedural equity are also taken into account (Breil et al. 2018).

## 6 Summary of results

In total, 22 measures were selected and analysed using the methods described above (Table 1). A majority of measures are technical or 'grey' measures. This may be explained by the familiarity of the stakeholders and end users with this type of measures. The governance analysis shows that the knowledge and administrative resources for implementation of these
measures are present at the sites, and implementation generally does not require the involvement of a broad range of stakeholders. Also, the effectiveness of these measures can often be modelled and is less uncertain than for instance behavioural measures. This is in line with Dhakal & Chevalier (2017) who find that, in the case of urban storm water management, technical solutions remain preferred throughout the world. However, the socio-economic analysis shows that these technical infrastructure measures are often expensive, particularly when compared with blue-green solutions or behavioural measures.

In the case of Badalona, the grey infrastructure proposed has the highest level of risk reduction, but is also much more expensive than the SUDS and the EWS. In fact, the cost benefit analysis shows that the investment and operational costs are not compensated by the socio-economic benefits considered. The proposed SUDS have a lower potential for flood and CSO risk reduction (also because the measures analysed only covered a small area of the city), but the improvements they bring for instance to habitat creation and enhanced aesthetic and recreational value (Locatelli et al., 2020), gives them a higher net
benefit. The EWS was the most cost-effective measure, significantly reducing flood risk.

When the measures are compared by their governance needs, we see a different picture. In Badalona, the measures that propose an increase of sewer capacity are part of an already existing Urban Drainage Master Plan. That means that the knowledge and competences to implement these measures is readily available. This is labelled by Dhakal and Chevalier (2017) as 'pro-grey arrangements'. The main barriers to implementation are funding, political decision making and disturbance to due to
construction works. While the SUDS require a smaller budget, there is limited experience on how to implement them and technical expertise and standards/guidelines are currently lacking, although relevant knowledge can be obtained from regional examples or local research partners. SUDS require the collaboration of a broad range of stakeholders, which requires coordination by the Badalona City Council. This makes implementation significantly more complicated than the proposed technical measures. These barriers are all acknowledged in other cases as well (Dhakal & Chevalier, 2017). The governance
needs for the EWS are mostly met, the main challenge is to develop and implement the required protocols for the response to the 'warnings' that the EWS gives.

In terms of social justice, as stated above, all measures have a positive impact on Badalona's citizens due to the decrease in the risk of floods and CSOs. The SUDS have an increased benefit, due to their many positive side effects, but some of them are local and depend on where the measures are implemented.

The case of Bergen shows similar results. The sewer separation (a traditional engineering measure), shows the highest potential for risk reduction, particularly in cases of extreme rainfall. However, they are also very expensive in relation to the risk

reduction achievable. The proposed SUDS measures are relatively low priced compared to their overall risk reduction potential, but do not have the potential to reduce the risk of extreme events. When combined with using the roads as safe flood ways (a clever way of repurposing the grey infrastructure), they are able to handle peak flows in urban drainage at lower costs than sewer separation. The combination of blue-green-grey measures has been proven successful in other studies as well (Alves et al., 2020; Depietri & McPhearson, 2017).

In the case of Bergen, all the governance needs for implementation of sewer separation are met. With regard to SUDS, there is still additional knowledge required on the performance of SUDS in cold climates. The BINGO project was instrumental in involving the required stakeholders and so meeting the organizational needs. However, there were too few incentives for private property owners to implement the required measures (see also Dhakal & Chevalier, 2017). This can be circumvented by first implementing the SUDS at municipally owned properties. The implementation of Safe Flood Ways is a less traditional technical solution. It adds a new functionality to roads that fall outside the responsibility of the road authorities and thus require coordination between different municipal authorities. An example of fragmented governance (Dhakal & Chevalier, 2017). Also, the broader impact on public safety when running water with high velocity through the streets needs to be assessed (Skrede, 2020).

All measures in Bergen benefit the general public, because of the reduction of the risk of CSOs. As in the Badalona case, the SUDS can provide many side benefits that have additional positive impacts. Negative side effects mostly involve construction and maintenance activities and resulting disturbances. Most measures are financed at the municipal level, reflecting the egalitarian or solidarity principle. SUDS or sewer separation implemented at private properties has to be financed privately, following the "polluter pays principle" (Strehl et al., 2019a)In the Veluwe case, the artificial retention measure, which involves constructing a large water transportation pipe from a nearby lake to the Veluwe, is the most expensive measure. It is also the measure with the highest potential for risk reduction, in this case measured as the additional groundwater recharge in the Veluwe groundwater system (approximately 30 $Mm^3$ meters per year). The additional recharge for the green measures (change in vegetation) ranges from 2-20 $Mm^3$ per year, but at much lower costs. Agricultural water restrictions are less expensive than the other measures. Most expenses go to helping farmers changes their farming practices (or buying them out), since water restrictions will force them to change crops. However, the amount of water saved is relatively small (0.2 – 0.3 $Mm^3$ per year).

At the Veluwe, the implementation of artificial retention is relatively easy, because the required knowledge is available and the required coordination between actors is limited and can be achieved through existing institutions. The implementation of land use change is much more complicated. It requires the collaboration between stakeholders outside of current arrangements and with diverging interests. The BINGO CoP has already been successful in establishing this cooperation. More importantly, changing land use has a huge impact on public opinion, since the Veluwe is well protected (Natura 2000) and a cherished spot for recreation. Changing its vegetation at the required scale would require a public debate on forest management at the national level (Van Alphen et al., 2019). Agricultural Water Restrictions require a locally embedded stakeholder process to be initiated,

involving famers, municipalities, water authorities and the Province. It requires farmers to change their crops and farming practices, which are often considered part of the cultural heritage as well.

Artificial infiltration improves the sustainability of the drinking water supply and helps preserving the groundwater system. These benefits are distributed equally among water users in the region, who, through fees, also bear the costs. The negative side effects are mostly decrease in attractiveness of the environment due to additional water (and energy) infrastructure. These negative effects impact disproportionally people who live nearby these infrastructures. Mitigation activities include minimization of visibility and ecological effects. Land use change has a number of positive effects (sustainable drinking water supply, preservation of groundwater system, a more diverse and robust landscape, increased biodiversity) that impact the general public. Cost for these measures are borne mostly by land owners, who will be compensated by either the province or through water fees. Negative effects mostly have to do with the loss of wild life and plants specific to pine forests (although overall bio-diversity will increase). During the transition period, tourism entrepreneurs may induce losses due to intensified foresting activities (Strehl et al, 2019a). For the measure Agricultural Water Restrictions, farmers affected by the measure would carry the major burden, but would be compensated by the regional or national government for loss of production capacity. The local groundwater supply and natural environment are positively affected which directly benefits land owners, local inhabitants and tourists.

In the case of Cyprus, the CEA shows that the most cost effective measure is the maintenance of groundwater recharge systems (in this case check dams), yielding a 1250 $m^3$ groundwater recharge per euro invested, compared to treated sewage water for irrigation (32,6 $m^3$ recycled water used per euro invested) , desalination (1,5 $m^3$ desalinated water consumed per euro invested) and irrigation scheduling technologies (0,90 $m^3$ water savings per euro invested). For the irrigation sector, irrigation scheduling technologies measure had the highest MCA weighting score (13.5) compared to the treated sewage water option (12.1). For the domestic water supply sector, groundwater recharge systems received the highest final MCA score (14.6) compared to the use of water desalination (13.3) (Strehl et al, 2019a).

According to the governance needs analysis, the implementation of this maintenance scheme mainly requires better coordination between the Water Authority and the local community councils. Structural, institutional and political rigidities negatively affect the adoption of irrigation scheduling technologies in Cyprus. The lack of political will to charge irrigators with water prices that cover the full costs, i.e., financial, environmental and resource, does not provide an incentive to invest in water saving technologies (Van Alphen et al. 2019). Giannakis et al. (2016) suggest that the low irrigation water price elasticities, the ageing and lower training levels of farming population, the small farm size and the low level of farm investments also impede the uptake of irrigation scheduling technologies. Support for farm training schemes, including issues such as water conservation and climate change adaptation, could improve the skills of the farmers and foster the adoption of new technologies (Giannakis and Bruggeman, 2015; 2018, Van Alphen et al., 2019).

It follows from the governance needs analysis that the use of treated sewage water for irrigation could be implemented relatively easily. However, the total benefit is small, considering that only 6% of the farmers have access to this source. Also, the long term effects of possible contaminants are yet unknown. For desalinization the key governance challenge is financial viability. Local households will pay a higher price for the desalinated water. Yet, as community councils will be responsible for selecting the source of water there are concerns regarding the prioritization of a cheaper source (Van Alphen et al. 2019).

All four measures proposed in the Cyprus Case Study are financed at least in part by the sectoral groups/communities that benefit directly and/or indirectly. Irrigation scheduling technologies and the maintenance of groundwater recharge systems have potential side effects which benefit the general public as they increase the qualitative and quantitative state of the groundwater system. Desalination and the use of treated sewage water for irrigation only benefit specific groups of water users, namely the  households of the downstream communities of Peristerona Watershed (desalination) and the farmers that have
access to the treated waste water. Also, they have notable negative side effects (impact of emerging contaminants, carbon emissions and brine discharges) which burden the general public and future generations (Strehl et al., 2019a)

In the first Wupper River Basin case (insufficient reservoir storage due to drought), it was found that the technical infrastructure measures are very expensive compared to the behavioural measures, also in relation to the level of risk reduction. The reduction of low water elevation (which effectively reduces the outflow from the reservoir) is by far the most cost effective measure
(€0.001/m$^3$). The water saving scheme is not expensive, but the effect on the capacity of the reservoir is relatively low, and so is the cost effectiveness (€0.194/m$^3$). Water transportation (€0.040/m$^3$) and the horizontal well (€0.054/m$^3$) rank in between, It was noted, however, that these infrastructural measures may be needed as a risk buffer in the future, since they provide redundant capacity.

With regard to the governance needs, the reduction of low water elevation is also the easiest measure to implement, with most
of the governance needs being met at the research site. The main concern is the effect on the downstream ecology when the outflow of the reservoir is reduced. The technical infrastructural measures in this case are much harder to implement. To build the transfer pipe between the two catchments, water authorities and the environmental agency should be involved, as well as the property owners affected by the route. It also requires setting legal standards and assessing the technical feasibility and environmental impact. The same goes for implementing a new abstraction well (horizontal well). An additional barrier for that
measure is the potential change in water quality (harder water) due to mixing of sources.

In terms of social justice, the measure Reduction of low water elevation enhances social justice by securing the water supply to the general public, without increasing the price of water. The main negative side effect as a reduced flow passing the dam, leading to a decrease in energy production and potential decline of ecological quality. This also affects the general public. The Substitution with an alternative water source, may increase existing inequalities since it increases the price of water, which
disproportionally impacts low income groups. The same goes for the Water transport between catchment, with the addition

that is negatively impacts the property owners near to the infrastructure. To the extent that these owners will be compensated by the Wupper Association, the cost will be carried by the general public. Finally, the pipe/channel route can have a negative impact on the environment and landscape, which impacts the general public.

For the second Wupper River Basin case (flood risk due to increased precipitation) three technical measures were analysed. The retention basin is the most expensive measure (88k€/y), but it also performs best in terms of risk reduction. Alignment protection (10k€/y) and protection measures for property (3k€/y) have a much smaller risk reduction effect, about 10 to 15 times smaller. Since these results are calculate for a specific scenario, it cannot be assumed that the just increasing the investment in the latter two protection measures will yield the same risk reduction as the retention basin.

According to the governance analysis, all requirements for implementing the retention basin are in place. With regard to the technical protection measure for property, one of the barriers for implementation is convincing the property owners to take action. Support and funding needs to be coordinated between public services and property owners. Flood protection is considered a public service instead of a (partially) private responsibility. When this is the case, it does not encourage private investment (Geaves & Penning-Rowsell, 2016). In the case of alignment protection, this is indeed a matter of public action, where land may be acquired from property owners, but no investment from their side is necessary.

From the social justice analysis, it follows that the retention basin will benefit people downstream of the basin, while property owners above the endangered areas have the basin built on or near their properties. The basin will be financed by the Wupper Association, but property owners may face decreasing value of their properties because of negative environmental impact or decreasing aesthetics. This can be mitigated by an appealing design and environmental friendly construction of the basin. A positive side effect of the basin that benefits the broader public is the improvement of water quality due to a reduction of direct 510 run offs into natural streams. Protection measures for property are generally paid for by the property owners, who also reap the benefits of reduced flood risk. In case of municipal buildings, the municipality has the opportunity to embellish public spaces by choosing an appealing design. The Alignment protection will most likely be financed by the Wupper Association, as the measure benefits the general public. In case property owners will bear the costs, this will likely lead to increased social inequalities.

In the Sorraia Valley in Portugal, the technical measures involve the rehabilitation and modernisation of existing irrigation networks, that consist of a canal, a transport and distribution system and a secondary irrigation system. Improving the canal is not the most cost effective in terms of cubic meters of water saved, but it is in terms of impacted area. Improving the transport and distribution system and the secondary irrigation system only affect a small areas and need to be replicated in other areas to reach the same impact as improving the canal (Strehl et al., 2019a). The Tagus Water Resources Management model has 520 the potential to be very cost effective, but this is dependent on the level of use the Water Authority will promote. This also the most important challenge with regards to the governance needs. Implementing the Water Resources Management model

requires a shift from a top-down management approach to a more network oriented governance model. This requires an integrated approach to water resource management and the participation of a broad range of actors. The rehabilitation and modernisation of the irrigation networks pose no specific governance challenges, apart from acquiring funding for the investments.

From the social justice analysis it follows that the rehabilitation and modernisation of irrigation networks mainly benefits the farmers, who also pay for the measures. To alleviate the financial burden, they can apply for funding. The assured agricultural sustainability in the region benefits a broader public as well. The Water Resource Management Model helps to better plan and manage water resources in the Tagus river basin, which benefits the general public. The costs are borne by the water authorities and then allocated to all water users through a tax or a fee.

## 7 Conclusion

The application of the BINGO approach has been successful in generating decision-relevant outcomes for developing adaptation strategies at the research sites. The governance analysis allowed to stakeholders to identify gaps in the governance needs to implement measures and to prepare steps to fill those gaps. The outcomes of the socio-economic analysis allowed stakeholders to prioritize measures by their cost-effectiveness, cost-benefit ratio or performance on a broader range of criteria. Sometimes this yielded surprising results, such as the high cost effectiveness of check dams maintenance in the case of Cyprus. Finally, the social justice analysis can help stakeholders choosing proper financing mechanism that fits the desired principle (solidarity, egalitarian, deontological) and gives a first indication of how positive and negative impacts are distributed over different groups. Although the research sites were very different, both in their challenges as well as their socio economic and institutional context, the approach presented in this paper yielded useful results in all cases. This supports the transferability of the approach to other cases in Europe.

However, we can identify specific benefits and limitations for each of the analyses (Table 4). The main benefit of the governance analysis is that is provides a systematic overview of the requirements for implementing a certain measure, with attention to a broad range of building blocks for adequate governance. This not limited to technical and economic aspects, but also includes cultural, communicative and legal aspects. A limitation in the way that the method was applied is that it does not provide specific thresholds for the required level of these indicators, other than reported by the researchers and stakeholders involved.

The socio-economic analysis contributed in structuring decision relevant information on adaptation measures focusing on potential outcomes of each measure. The methods applied help to quantify and/or rank indicators affecting costs and benefits of the selected measures, from a socio-economic point of view. Moreover, the methods can be integrated in a broader, scenario-

based approach to assessing adaptation strategies. Limitations of the method primarily deal with the availability of data, which has a strong effect on the validity and reliability of the conclusions drawn from the analysis.

Finally, the social justice analysis gives a broader perspective than the plain focus on the outcomes of adaptation and also considers the distributional effect on different groups in society. This may result in the identification of unbalanced burdens 555 or co-benefits which leads to better informed decisions and helps to realise climate justice. However, in the way the method was applied, the acquisition of meaningful social-justice information and derived interpretations relevant for decision makers, highly relies on the interview partners. They need to have a specific knowledge of the local adaptation measures/options planned, and the socio-economic environment.

**Table 4: Assessment of the applied analyses**

| BINGO analysis | Benefit | Limitation |
|---|---|---|
| Governance analysis | Provides systematic overview of requirements and whether they are met; takes into account broad range of factors, not only finances and technical capability. | Method itself does not provide standards in whether requirements are sufficiently met; relies on self-reporting by researchers and stakeholders. |
| Socio-economic analysis | Helps to structure decision-relevant information about adaptation alternatives, focusing on measurable outcomes of each option; applied science offers straightforward methods to quantify or at least rank relevant indicators affecting costs and benefits from a socio economic point of view; methods for a socio-economic analysis are flexible to integrate the scenario based thinking of climate change projections. | Limitations arise with data availability; in cases with very broad decision-relevant socio-economic indicators to cover, (un)reliable input data for a quantitative analysis effects the robustness of conclusions drawn from the analysis. |
| Social-justice analysis | Helps to focus not only on plain outcomes of adaptation, but also on | Information acquisition for a social-justice analysis relies on qualitative input, e.g. by |

| | distributional effects among society; broadens the scope of the analysis, eventually leading to identify additional co-benefits or unbalanced burdens for stakeholders of climate change adaptation measures, allowing a better informed decision. | interviews and pre-structured questionnaires as conducted in the BINGO-project; time and financial resources and available interview partners may limit the scope of the analysis. |


**Code and data availability.** Model files and data are not provided due to the confidentiality of the data and models. Notwithstanding, in agreement with the other project stakeholders, the authors of this paper will try to address specific requests for scientific purposes.

**Author contributions:** HJA, CS, FV, EI, AP, SG and EB developed the methodology. All authors were involved in the
research at the research sites. HJA, CS, FV, EI, AP and SG prepared the paper with contributions from all authors.

**Competing interests.** The authors declare that they have no conflict of interest.

**Acknowledgments:** The authors wish to thank all partners and stakeholders of BINGO for their input and feedback. We specifically thank Adriana Bruggeman, Christos Zoumides, Hakan Djuma, Marinos Eliades (The Cyprus Institute); Ayis Iacovides, Marios Mouskoundis (IACO); Maria Rafaela de Saldanha Gonçalves Matos, Ana Estela Barbosa, Maria João
Freitas, Teresa Viseu (LNEC); Alberto Freitas (DGADR); Ana Luís (EPAL); Eduardo Martinez-Gomariz (Cetaqua); Beniamino Russo (Aquatec); Rita Andrade (SPI); Thorsten Luckner, Paula Lorza (Wupperverband); Suzanne Buil-van den Bos (Provincie Gelderland); Jan Hogendoorn, Jolijn van Engelenburg (Vitens); Juliane Koti, Andreas Hein, Leni Handelsmann (IWW); Pedro Brito (DGADR); Robert Mittelstädt (Hydrotec); Marit Aase, Magnar Sekse (Municipality of Bergen); Ashenafi Seifu Grange, Tone Merete Muthanna (NTNU); Adriana Hulsmann; Nicolien van Aalderen (KWR)

**Financial support**. This research has been supported by the BINGO European H2020 project (grant no. 641739).

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
