# Peer review of "Selecting and analysing climate change adaptation measures at six research sites across Europe"

_Natural Hazards and Earth System Sciences, 2020_

## Referee Comment (RC1) · Anonymous Referee #1 · 16 Jul 2020

The paper contains the results of an development and assessment of methods and tools in relation to climate change adaptation across several sites in Europe. As such it is of interest to the readers of NHESS. The paper has been submitted to a special issue that puts emphasis on case studies, stakeholder engagement and practical applicability. I have taken that into consideration when doing the review by focussing on the take-home messages that other locations could benefit from receiving.

I would appreciate more consistant and more frequent citation of existing literature and published project outcomes. Some examples from section 2: in line 57-58 two databases are mentioned and a method. Also the CoP should be defined here (if

not in the introduction), both by a brief explanation and a citation. In line 66 a tool should be referenced as well (I assume that the restrictions to accessibility in line 430 does not apply to the tool). Also ensure that the references to the cited literature are complete and searchable by providing URL, ISBN or similar, the reference list really needs improvement.

The novelty of the tool in section 2 should be described: what is new, are the headings and structure similar to the databases or adapted as an outcome of the study/workshops/project? There seems to be two dimensions in the tool (lines 67-69 versus 73-75), is that correctly understood? Please clarify and extend this section, perhaps inspired by the content of section 3.1, that is much easier to follow.

I think I agree with the statements in section 4.1, but I have difficulties following the argumentation and what the practical outcome of reading the section should be for the reader? Please consider rewording. The paper by Markanday et al (2019) is clear about which shortcomings they think exist and what should be done, perhaps that can serve as an inspiration. In section 4.2 it would be nice to state that while Figure 1 helps in deciding which framework is most relevant, each of the frameworks are known to only consider quantitative metrics, while in reality decision-making is a more complex process (as is also discussed in the section). Some of the questions that are in this paper discussed under the heading of social justice is part of a full Cost-Benefit Analysis (e.g. Boardman et al 2018: Cost-Benefit Analysis. Concepts and Practice. https://www.cambridge.org/core/books/costbenefit-analysis/484720E57798B7E7A29C7156407CD4A1 )

Overall the description of the economic frameworks in section 4.3-4.6 takes up a lot of space considering it is relatively standard textbook material. I would recommend to abbreviate this a lot and rather focus on e.g. the stakeholder processes or the selection of adaptation measures.

L 349: Using decadal predictions of 5 years seem to be a contradiction of terms? Is it a

writing mistake or is the underlying assumption of the BINGO project to only consider the current climate in climate adaptation projects ? Please clarify/justify.

The discussion section contains a very strong message about the shortcomings of technical infrastructure. My own experience is that some stakeholders are very much in favour of such solutions while other stakeholders are very much against them. If the BINGO project has been able to overcome this shortcoming it would be of interest to know how this was done and which of the tools discussed in the previous sections of the paper were particularly useful in obtaining this result. As it stands now the discussion section seems to report more on the belief of the authors than the result of the analysis.

Overall it is difficult to keep track of what is sensible solutions in relation to the different sites. One large table combining the information from Tables 1-3 and also supplementing with current and future climate change states and challenges (not only climate risk) in the beginning of the paper would be a help to the reader and also in line with the requirements of the purpose of the special issue. It would also serve as a basis for supporting the analysis supplied as a result of the study / project in Table 4. Why does Table 4 not contain information about the adaptation measures, it would seem logical to summarize all sections in this table?

Minor points:

Figure 1: The text on mandatory basis could benefit from being included into the legend. I am surprised that future states given the adaptation measures are not needed in accordance with e.g. Zhou et al (2012). If it is because changes after 2025 is not considered, then it should be justified. The economic tools shown in Figure 1 all require more than five years of input to the economic analysis.

L139: I suggest to refer to a revised section 2.

L141: What "part 1" are you referring to?

L167: The homepage probably also contain other relevant resources and should be moved to a more suitable place in the paper.

The paper would benefit from being proof-read by a native English speaking person.

---

## Short Comment (SC1) · 10 Aug 2020

Dear colleague, thanks for you thorough review of our paper, which contains useful suggestions to improve it. I will discuss them with my co-authors and update the paper accordingly. Best regards, Henk-Jan van Alphen
* * *

---

## Short Comment (SC2) · 28 Aug 2020

The paper provides a valuable framework on how to select and evaluate appropriate climate change adaptation measures. This is of high importance and can assist cities to decide on the most fitting, most effective, least costly and/or most socially just measure. As cities are currently investing into adaptation measures, the BINGO results provide guidance in the right moment in order to avoid miss-investments. I would have expected more mentioning of similar research projects, reports on green blue infrastructure and citation of publications. I was missing an explanation on how the preselection of measures is to be conducted. This should at least be coarsely described.

[Figure]

It is only mentioned in line 60, that it is based on hazard and risk identification analyses. Instead, the chapter on economic frameworks is very detailed and can potentially be shortened a bit. Also, I was missing the mentioning and a short presentation of the case studies in the beginning. Further, I suggest to combine tables 1, 2 and 3. The arrows in figure 1 could be enlarged a bit. Line 141 refers to "part 1", which is not provided. 142: Please explain a bit the "risk assessment" that is mentioned here. Line 199 talks about "indicators" a bit out of the blue, please elaborate a bit on indicators used for the assessments. Line 201-203 contains the word "simulating" 3 times. Line 223 mentions "criteria" which could also be called "indicators". Line 262 uses the word "parameters" – is this the same as "criteria" or "indicators"? Please make sure that consistent wording is applied. Line 264: Not clear between what the pairwise comparisons are to be conducted. Line 267: What is an AHP analysis? Line 366: First mentioning of "CSOs" – provide full word. In the Discussion I would find sub-headdings quite helpful, e.g. "6.1. technical infrastructure measures", "6.2. blue/green solutions" and "6.3. behavioral measures". Line 399: Do you mean "water is too cheap" – instead of "cheap water"? Line 403-404: The description of "Reduction of low water elevation" should be provided earlier in the manuscript. This measure is mentioned several times earlier, but the explanation of what is meant with this measure, comes in the very end. Pease also consider that such a decision has to be agreed to by authorities in order to ensure that the minimal ecologically required water flow in the downstream river sections is ensured. Line 414: Instead of "values" maybe "tresholds/aims" would be a more appropriate term. Table 4: Name "Benefits and limitations of the applied analysis" as caption.

---

## Referee Comment (RC2) · Anonymous Referee #2 · 18 Sep 2020

The paper describes three types of analysis that can support selection of adaptation measures. The analyses have been applied to six research cases across Europe. Overall the paper provides practical insights in the many considerations that need to be made in such a decision making process on what benefits to incorporate in the analysis, how to value or rate them, what implementation barriers to consider (and what this means for the inclusion of stakeholders) and how to assess potential increase in injustice as a result of adaptation Altogether this is a big story that has been addressed, mostly on its separate components, extensively in literature. It is difficult to find out what are the new elements this paper tries to add to the already existing body of literature. If it does add new elements they are difficult to distinguish as the paper fails to

pose clear research questions and structure the outcomes accordingly. The number of analyses: governance, socio-economic and social justice is too much, too generic and for a large part common practice to arrive at clear new conclusions. The paper would benefit if a clear choice would have been made for one or two of those analyses with a focus on new elements. To show the reader what is new, a better and more structural assessment of existing literature would be needed: what is the current status quo what is BINGO adding? Social justice could be this new element and a good methodology to introduce this into adaptation decision making could be very useful. The current short description and consecutive analysis are not yet convincing. It may give the impression that social justice analysis for most common adaptation solutions is obsolete. In its presentation the paper can be improved a lot by more concise and precise language. A review by a native speaker is recommended. In conclusion I think the paper needs rethinking of the key questions and messages, selection of topics and restructuring. A journal that is more focused on governance and implementation issues of adaptation seems more appropriate. A few comments by section: Abstract: the main conclusion that decision relevant outcomes have been achieved is not made clear by the results. Research questions are missing. 3.1 Unclear what improvements have been made to the 3-layer framework 4.1 Challenges are posed but not sufficiently explained and supported by references 4.2 Many of these MCA/CE/CBA frameworks have been proposed in previous EU research projects such as ECONADAPT and BASE and published. 5.3 should be 'the application of social justice Discussion: this section does not systematically discuss what has been presented in previous sections but presents a whole suite of new observations from the cases that are insightful however lack structure and coherence with the rest of the paper. cumbersome language examples r105 'analysis of the assessment by analysis ... r119 This issue is also linked to, r125 so called toolbox, r153 Integration of stakeholders..

---

## Author Comment (AC1) · 29 Oct 2020

Thank you for reading our paper an providing thoughtful comments. We will try to address your comments in this response and of course in the possible revision of the paper.

We will include more references to existing literature and published project outcomes, as well as improve the reference list so that it is complete and searchable.

With regard to the tool mentioned in section 2, we will include a more detailed description of the tool, its structure and its relation to the databases.

[Figure]

Section 4.1, which you refer to, is just meant as an introduction to the challenges in analysing socio-economic implications – and the bridge to the answers and guidance to overcome these challenges. Details are given in our paper in the subsequent chapters. We will write an additional sentence to clarify this at the end of section 4.1.

For section 4.2 (figure 1), we believe that MCA, but also CBA and CEA can be designed to also include qualitative elements (e.g. "full" CBA). We suggest to add "full" CBA to the CBA cell in figure 1. We agree that social justice can be part of a full CBA, but since we did not include it in the CBA we also describe it separately in this paper.

We will abbreviate the description of the economic frameworks as you suggest with regard to section 4.3-4.6.

With regard to apparent contradiction of the decadal predictions for 2025. The BINGO-project started in 2015, which is also the starting point for the decadal predictions, hence the time horizon of 2025. We will clarify this in the text.

We agree that it is not always possible to trace the elements of the discussion to the analysis, which can be improved by making stronger links between the discussion and the preceding parts. We kindly accept your suggestion to merge and supplementing the tables and move them (in part) to the introduction.

With regard to your comment on the inclusion of future states in the economic analysis: the cases in general covered a range of decadal predictions (2015-2025) and their effects on e.g. raw water availability. Thus, a potential range of changes is covered. Costs are usually based on the lifetime of a measure, calculated either as annual costs (annuity method, dynamic cost calculation) or as present value of costs (also dynamic cost calculation). Thus, economics cover a plausible time horizon. We will expand the manuscript for the final draft with explaining sentences.

We hope we have covered the main points to your satisfaction and want to thank you again for your thoughtful review.

---

## Author Comment (AC2) · 29 Oct 2020

Thank you for attention to and appreciation of our paper and your thoughtful comments. We will address the comments below and in a possible revision of the paper.

We will include more references to literature, similar studies and reports on blue green infrastructure.

With regard to the preselection of measures, we will extend the description of that process, potentially abbreviating the description of economic frameworks.

We will, also in response to the first referee include a table with the research sites

and their main climate risks in the introduction, as well as a brief description of each research site. We will merge tables 1-3 and enlarge the arrows in figure 1.

The reference to "part 1" is a mistake and should be "section 2". This will be changed in the final draft. We will also provide an explanation of the "risk assessment".

We will elaborate on the "pairwise comparison" and the "indicators", use consistent language in referring to them and make the section easier to read by improving the language.

We will check the manuscript for any acronyms that have not been introduced in their full form, add subheadings to the discussion and give a brief description of the measures that are mentioned in the text.

We agree that thresholds is a more appropriate term than values and kindly accept the suggested change of the table caption.

We hope we have covered the main points to your satisfaction and want to thank you again for your thoughtful review.

---

## Author Comment (AC3) · 29 Oct 2020

Thank you for reading our paper an providing thoughtful comments. We will try to address your comments in this response and of course in the possible revision of the paper.

You raise a number of fundamental points with regard to the structure of the paper, the topics it covers and to what extent it adds new insights to the existing literature. We believe that the approach of combining different analyses and conducting them in co-creation with stakeholders in quite different circumstances and case studies is what the work in BINGO adds. We will try to accentuate that, also by providing a more structural

review of existing literature, as you suggest and give more explanation of why we did the analyses and what the outcomes were. We will also improve the abstract likewise.

The social justice analysis has its limitations, we agree, and these are mentioned in the text. In spite of these limitations, we found it helpful to confront the stakeholders/decision makers with the topic. We agree that it could be improved, also regarding the methodological side of the approach, but this exceeded the scope of BINGO as well as of this article. We will mention the potential of such an analysis, and that it needs more work/applications.

With regard to section 3.1, we will better explain what adaptation were made to the 3-layer framework.

With regard to the challenges mentioned in section 4.1: we included the reference to a recent peer review paper with a literature review of the topic as well as a conference paper with a literature review about it (Markanday et al., 2019; Dogulu and Kentel, 2015). One challenge briefly noted in 4.1 is the difficulty to bring together different stakeholders and different data types and knowledge about it in a methodology-format best fitting for a case. We will make that more explicit in the text of 4.1.

With regard to section 4.2, we agree that a growing number of projects dealing with climate change adaptation and methods tested exist, which we reckon a good development! MCA/CE/CBA frameworks have been published before, but their scientific base used and guidance for climate change adaptation are still developing. Additionally we have coupled the frameworks with risk assessment and also investigated the benefit of method-combinations (please see our section 4.7 combining frameworks).

We will change the heading of 5.3 as to your suggestion.

The discussion can indeed be improved by providing better links to the main text. Also, we will improve the language so that it is less cumbersome to read.

We hope we have covered the main points to your satisfaction and want to thank you

again for your thoughtful review.

---

## Author Response (AR1)

**Point-by-point response to the editor and referees**

This document contains a point-by-point response to the comments of the referees. If relevant, row numbers are given to show where changes in the manuscript have been made.

We would like to thank the editor and referees for their thoughtful comments and we hope we have addressed them to their satisfaction.

**Editor comments**

Dear Authors, thank you for your considerate responses to the review. The article will need major revisions and will be re-reviewed.

EC#1: Clarify the novelty of your work and its contribution to societal responses to natural hazards. Make efficient use of recent literature on the topic.

We have provided a clearer statement of the purpose of the article and clarified the main contribution (combining different analyses in a participatory framework) by expanding the review of literature in the introduction (r35-91)

EC#2: Improve the tables, as suggested by reviewer 1. Make sure to tap in to the diverse knowledge and experience of all co-authors.

We have integrated tables 1-3 in one table (table 1) including climate change risks linked to specific future hazards (increased precipitation or drought).

**Referee #1**

The paper contains the results of an development and assessment of methods and tools in relation to climate change adaptation across several sites in Europe. As such it is of interest to the readers of NHESS. The paper has been submitted to a special issue that puts emphasis on case studies, stakeholder engagement and practical applicability. I have taken that into consideration when doing the review by focussing on the take-home messages that other locations could benefit from receiving.

C#1: I would appreciate more consistent and more frequent citation of existing literature and published project outcomes. Some examples from section 2: in line 57-58 two databases are mentioned and a method.

We have added a short literature review to the introduction (r37-57) and added citations throughout the text.

C#2: Also the CoP should be defined here (if not in the introduction), both by a brief explanation and a citation.

We have added a short description of the CoP, including a reference (r76-78)

C#3: In line 66 a tool should be referenced as well (I assume that the restrictions to accessibility in line 430 does not apply to the tool).

The tool is now referenced (moved to r104)

C#4: Also ensure that the references to the cited literature are complete and searchable by providing URL, ISBN or similar, the reference list really needs improvement.

The reference list has been updated, all references now contain a URL and ISBN as far as we could obtain them.

C#5: The novelty of the tool in section 2 should be described: what is new, are the headings and structure similar to the databases or adapted as an outcome of the study/workshops/project?

We have expanded the description of the tool, better explaining its structure in relation to the project. (r105-114)

C#6: There seems to be two dimensions in the tool (lines 67- 69 versus 73-75), is that correctly understood? Please clarify and extend this section, perhaps inspired by the content of section 3.1, that is much easier to follow.

The tool is now described in r105-114. The governance aspects listed in r118-119 (previously 73-75) are not part of the tool, but were discussed in the CoPs as part of the measures' selection process.

C#7: I think I agree with the statements in section 4.1, but I have difficulties following the argumentation and what the practical outcome of reading the section should be for the reader? Please consider rewording. The paper by Markanday et al (2019) is clear about which shortcomings they think exist and what should be done, perhaps that can serve as an inspiration.

We have decided to remove section 4.1 (sections have been renamed in the new draft).

C#8: In section 4.2 it would be nice to state that while Figure 1 helps in deciding which framework is most relevant, each of the frameworks are known to only consider quantitative metrics, while in reality decision making is a more complex process (as is also discussed in the section).

We agree it would be helpful to give advice in terms of which framework is most relevant. But that is hardly possible, because it is very case dependent. The decision tree gives an idea which framework might be useful for a specific case, by answering the questions along. It leads the reader to that framework, potentially best fitting for the reader's specific case. In addition please note: non-monetary values (relating to the first question in the decision tree) can also be non-quantitative at first sight. There are MCA methods helping to structure stakeholders preferences (at first sight non-quantitative) and translate them into (at least) "proxy" values, that can be used in a calculation. This is for example done in the method called analytical hierarchy process. A method which we also used in a modified way in one case study in BINGO.

C#9: Some of the questions that are in this paper discussed under the heading of social justice is part of a full Cost-Benefit Analysis (e.g. Boardman et al 2018: Cost-Benefit Analysis. Concepts and Practice. https://www.cambridge.org/core/books/costbenefitanalysis/484720E57798B7E7A29C7156407CD4A1 )

We agree, and this depends on how to design/frame the CBA. We did not include social justice elements in our CBA, that is why we kept it separate.

C#10: Overall the description of the economic frameworks in section 4.3-4.6 takes up a lot of space considering it is relatively standard textbook material. I would recommend to abbreviate this a lot and rather focus on e.g. the stakeholder processes or the selection of adaptation measures.

We have substantially abbreviated the economic frameworks section (section 4).

C#11: L 349: Using decadal predictions of 5 years seem to be a contradiction of terms? Is it a writing mistake or is the underlying assumption of the BINGO project to only consider the current climate in climate adaptation projects ? Please clarify/justify.

The decade starts at the beginning of the project (2015) while the results presented in this article are from a later date. The decadal predictions were also developed at the start of the project. However, we have rewritten this entire section and removed the text about decadal predictions.

C#12: The discussion section contains a very strong message about the shortcomings of technical infrastructure. My own experience is that some stakeholders are very much in favour of such solutions while other stakeholders are very much against them. If the BINGO project has been able to overcome this shortcoming it would be of interest to know how this was done and which of the tools discussed in the previous sections of the paper were particularly useful in obtaining this result. As it stands now the discussion section seems to report more on the belief of the authors than the result of the analysis. Overall it is difficult to keep track of what is sensible solutions in relation to the different sites.

We have completely rewritten section 6, which now contains a summary of the results of the three analyses and is renamed accordingly (r356-517)

C#13: One large table combining the information from Tables 1-3 and also supplementing with current and future climate change states and challenges (not only climate risk) in the beginning of the paper would be a help to the reader and also in line with the requirements of the purpose of the special issue. It would also serve as a basis for supporting the analysis supplied as a result of the study / project in Table 4.

We have integrated tables 1-3 in one table (table 1) including climate change risks linked to specific future hazards (increased precipitation or drought).

C#14: Why does Table 4 not contain information about the adaptation measures, it would seem logical to summarize all sections in this table?

Table 4 is meant to be specifically about the pros and cons of the applied analyses, not about the outcomes of the analyses.

Minor points:

C#15: Figure 1: The text on mandatory basis could benefit from being included into the legend. I am surprised that future states given the adaptation measures are not needed in accordance with e.g. Zhou et al (2012). If it is because changes after 2025 is not considered, then it should be justified. The economic tools shown in Figure 1 all require more than five years of input to the economic analysis.

Many thanks for the valuable comment. The cases in general covered a range of decadal predictions and their effects on e.g. raw water availability. Thus, a potential range of changes is covered. Costs are usually based on the lifetime of a measure, calculated either as annual costs (annuity method, dynamic cost calculation) or as present value of costs (also dynamic cost calculation). Thus, economics cover a plausible time horizon. We have expanded the manuscript with explaining sentences (r186-192)

C#16: L139: I suggest to refer to a revised section 2. & C#17: L141: What "part 1" are you referring to?

We have changed this part and we think a reference is not relevant anymore (r183-186)

C#18: L167: The homepage probably also contain other relevant resources and should be moved to a more suitable place in the paper.

We have moved the homepage link to the introduction (r84).

C#19: The paper would benefit from being proof-read by a native English speaking person.

We have improved the language throughout the paper.

**Referee #2**

The paper provides a valuable framework on how to select and evaluate appropriate climate change adaptation measures. This is of high importance and can assist cities to decide on the most fitting, most effective, least costly and/or most socially just measure. As cities are currently investing into adaptation measures, the BINGO results provide guidance in the right moment in order to avoid miss-investments.

C#20: I would have expected more mentioning of similar research projects, reports on green blue infrastructure and citation of publications.

We have added a short literature review to the introduction (r37-57) and added citations throughout the text.

C#21: I was missing an explanation on how the preselection of measures is to be conducted. This should at least be coarsely described. It is only mentioned in line 60, that it is based on hazard and risk identification analyses. Instead, the chapter on economic frameworks is very detailed and can potentially be shortened a bit.

We have expanded the section on measure selection (r92-124)

C#22: Also, I was missing the mentioning and a short presentation of the case studies in the beginning.

We have added a short description of the case studies to the introduction (r64-71)

C#23: Further, I suggest to combine tables 1, 2 and 3. The arrows in figure 1 could be enlarged a bit.

We have combined the tables in one table (table 1) and enlarged the arrows in figure 1.

C#24: Line 141 refers to "part 1", which is not provided.

We have changed the text and removed the reference.

C#25: 142: Please explain a bit the "risk assessment" that is mentioned here.

We have added an explanation (r185-192)

C#26: Line 199 talks about "indicators" a bit out of the blue, please elaborate a bit on indicators used for the assessments.

The indicators are highly case relevant, so for each case different indicators were chosen. We have reworded the sentence to reflect this (r235)

C#27: Line 201-203 contains the word "simulating" 3 times.

This has been changed

C#28: Line 223 mentions "criteria" which could also be called "indicators".

This has been changed

C#29: Line 262 uses the word "parameters" – is this the same as "criteria" or "indicators"? Please make sure that consistent wording is applied.

This has been all changed to indicators

C#30: Line 264: Not clear between what the pairwise comparisons are to be conducted.

The pairwise comparisons are done to find out which criteria out of two is of more importance than the other to a stakeholder. This, done for a set of indicators, having a pairwise comparison for each "one on one" comparison, helps to define a weighting for each indicator individually. Still, this individual weight aggregates the results of all pairwise comparisons of a specific criteria to all others. We have rephrased these lines to make this clearer (289-291)

C#31: Line 267: What is an AHP analysis?

AHP: Analytic Hierarchy Process, a multi-criteria decision support method. Reference is given in the manuscript for readers interested in more details (Saaty publications) (r291)

C#32: Line 366: First mentioning of "CSOs" – provide full word.

We have checked all acronyms and provide full words at first mentioning

C#33: In the Discussion I would find sub-headings quite helpful, e.g. "6.1. technical infrastructure measures", "6.2. blue/green solutions" and "6.3. behavioral measures".

We have completely rewritten section 6, which now contains a summary of the results of the three analyses and is renamed accordingly (r356-517)

C#34: Line 399: Do you mean "water is too cheap" – instead of "cheap water"?

Text removed

C#35: Line 403-404: The description of "Reduction of low water elevation" should be provided earlier in the manuscript. This measure is mentioned several times earlier, but the explanation of what is meant with this measure, comes in the very end. Pease also consider that such a decision has to be agreed to by authorities in order to ensure that the minimal ecologically required water flow in the downstream river sections is ensured.

The measure is now introduced in table 1 and in the text at r461 with a brief explanation in brackets.

C#36: Line 414: Instead of "values" maybe "tresholds/aims" would be a more appropriate term. Table 4: Name "Benefits and limitations of the applied analysis" as caption.

Has been changed (r528)

**Referee #3**

The paper describes three types of analysis that can support selection of adaptation measures. The analyses have been applied to six research cases across Europe. Overall the paper provides practical insights in the many considerations that need to be made in such a decision making process on what benefits to incorporate in the analysis, how to value or rate them, what implementation barriers to consider (and what this means for the inclusion of stakeholders) and how to assess potential increase in injustice as a result of adaptation

C#37: Altogether this is a big story that has been addressed, mostly on its separate components, extensively in literature. It is difficult to find out what are the new elements this paper tries to add to the already existing body of literature. If it does add new elements they are difficult to distinguish as the paper fails to pose clear research questions and structure the outcomes accordingly.

We have added a short literature review to the introduction (r37-57) and added citations throughout the text. We have provided a clearer statement of the purpose of the article in the introduction (r59-64) and we have rewritten section 6 to give a summary of results.

C#38: The number of analyses: governance, socio-economic and social justice is too much, too generic and for a large part common practice to arrive at clear new conclusions. The paper would benefit if a clear choice would have been made for one or two of those analyses with a focus on new elements. To show the reader what is new, a better and more structural assessment of existing literature would be needed: what is the current status quo what is BINGO adding?

We have added a short literature review to the introduction (r37-57) and added citations throughout the text. Also, we have rewritten the "discussion section" to now include a summary of the results of the analyses.

C#39: Social justice could be this new element and a good methodology to introduce this into adaptation decision making could be very useful. The current short description and consecutive analysis are not yet convincing. It may give the impression that social justice analysis for most common adaptation solutions is obsolete.

The social justice analysis has its limitations, we agree, and these are mentioned in the text. In spite of these limitations, we found it helpful to confront the stakeholders/decision makers with the topic. We agree that it could be improved, also regarding the methodological side of the approach, but this exceeded the scope of BINGO as well as of this article. We mention the potential of such an analysis, and that it needs more work/applications. We also provide more results of the case studies in section 6.

C#40: In its presentation the paper can be improved a lot by more concise and precise language. A review by a native speaker is recommended.

We have improved the language throughout the article.

C#41: In conclusion I think the paper needs rethinking of the key questions and messages, selection of topics and restructuring. A journal that is more focused on governance and implementation issues of adaptation seems more appropriate.

We have provided a clearer statement of the purpose of the article in the introduction (r59-64) and we have rewritten section 6 to give a summary of results. We also shortened section 4 substantially and provided more information on the case studies in the introduction.

A few comments by section:

C#42: Abstract: the main conclusion that decision relevant outcomes have been achieved is not made clear by the results. Research questions are missing.

We have provided a clearer statement of the purpose of the article in the abstract (r24-27). In the conclusion we have highlighted how the outcomes can be relevant to decision making, supporting the statement in the abstract (r32-34).

C#43: 3.1 Unclear what improvements have been made to the 3-layer framework

No improvement to the framework was made, but we made it into a questionnaire which focuses on individual adaptation measures. We have expanded the section on the 3-layer framework describing the questionnaire (r145-149).

C#44: 4.1 Challenges are posed but not sufficiently explained and supported by references

The section has been removed and its main message is now included under the former 4.2-section.

C#45: 4.2 Many of these MCA/CE/CBA frameworks have been proposed in previous EU research projects such as ECONADAPT and BASE and published.

We agree with the reviewer. Projects dealing with climate change adaptation and methods tested are growing, which we reckon a good development! MCA/CE/CBA frameworks have been published before, but their scientific based use and guidance for climate change adaptation are still developing. Additionally we have coupled the frameworks with risk assessment and also investigated the benefit of method-combinations (please see our section 4.6 combining frameworks). We have included a reference to the ECONADAPT and BASE projects.

C#46: 5.3 should be 'the application of social justice

Has been changed

C#47: Discussion: this section does not systematically discuss what has been presented in previous sections but presents a whole suite of new observations from the cases that are insightful however lack structure and coherence with the rest of the paper.

We have completely rewritten section 6, which now contains a summary of the results of the three analyses and is renamed accordingly (r356-517)

C#48: Cumbersome language examples r105 'analysis of the assessment by analysis ... r119 This issue is also linked to, r125 so called toolbox, r153 Integration of stakeholders..

All examples above (and others) have been changed/improved

---

## Author Response (AR2)

Dear Editor,

Thank you for your thorough review of our revised draft. We have addressed your comments in the new draft, please find a point by point response below (lines refer to the marked-up manuscript).

l.75: municipalities or communities (l.495)?

*Must be communities, we have changed the wording (l.71)*

l.82: Please give an explanation and reference from the peer-reviewed literature for Communities of Practice.

*We have added two references with a brief explanation (l.81-84)*

l.108-110: Please add this sentence to the previous paragraph (2 approaches).

*We have done so (l.105-108)*

l.110-124: It is not clear how many measures in the online portfolio are from the desk study, how many are new and/or how many are a combination (specific implementation). How (and by whom) were the governance needs of the existing measures from the desk study assessed? Isn't a governance assessment case specific?

*We have provided more explanation about how many measures were compiled in each step and how we reached the selection for the online portfolio. (l.110-113)*

*We have assessed the governance needs for the measures in the portfolio on a general level (not site specific) (l.124).*

l.135 Table 1: Considering the advantage of the flexibility of the socio-economic approach, as mentioned in the response, and in light of the added descriptions of the cases studies in Section 6, please add here the socio-economic method (MCA, CBA..) used for each measure or case study (and refer to figure 1). Consider making the research site a subheader, instead of a column, to keep the overview clear.

*We have changed Table 1 according to your suggestion.*

l.157: Please add the questionnaire as supplemental information.

*We have added the questionnaire as supplemental information*

l.278: anno –> year

*We have changed the wording (l.249)*

l.418-489: Here you mix analysis methods and case studies in a not always easy-to-follow manner, sometimes 1 method 3 case studies, sometimes 1 case study 2 methods. Please re-organize a bit.

*We have reorganized the text so that each part follows the same structure. (l.383-453)*

l.419: Socio-economic and governance -> Socio-economic

*We have changed the wording (l.375)*

l.435-439: Mm3 (as elsewhere in the manuscript)

*We have changed all instances in the text*

l.441: propose an increase

*We have changed the wording (l.384)*

l.490-514: Please clarify what analysis methods were used to elucidate all this information.

*We have added CEA and MCA to the description. (l.476-483) This can now also be found in Table 1.*

l.495: Water Authority?

*We have changed the wording (l.484)*

l.513: Not clear what specific groups.

*We have added the specific groups (households of the downstream communities of Peristerona Watershed and farmers that have access to treated waste water (l.501-502)*

l.634-638: Derived from socio-economics or social justice or both?

*Derived from social justice, we have added that to the text (l.558)*

l.641-644: Please stay with same order (governance before socio-economics)

*We have changed the order (l.565-570)*

Again, many thanks for the review and we hope to have addressed these comments to your satisfaction.